# Frequency and perceived influence of menopausal symptoms on training and performance in female endurance athletes

Heather M. Hamilton[1]*, Natalie M. Yarish[2], Kristin E. Heron[3]

1 School of Rehabilitation Sciences, Ellmer College of Health Sciences, Old Dominion University, Norfolk, Virginia, United States of America, 2 School of Exercise Science, Ellmer College of Health Sciences, Old Dominion University, Norfolk, Virginia, United States of America, 3 Department of Psychology, Old Dominion University, Norfolk, Virginia, United States of America

* hmcconch@odu.edu

## Abstract

Menopause is associated with several negative health concerns and common menopausal symptoms. Physical activity is essential to mitigate these negative effects, but menopausal symptoms may interfere with participation in physical activity. The purpose of this study was to determine the frequency and the perceived negative effect of menopausal symptoms among female endurance athletes. Female runners, cyclists, swimmers, and triathletes 40–60 years of age were recruited ($N = 187$). Participants completed an online, anonymous survey that included self-reported menopausal status; the Menopausal Rating Scale (MRS), a validated measure for assessing menopausal symptoms; and the perceived effect of menopausal symptoms on training and performance. Kruskal-Wallis tests were used to examine differences in total MRS scores among participants reporting no, slight, moderate, and strong negative effects of symptoms on training and performance. The most commonly reported menopausal symptoms were sleep problems (88%), physical and mental exhaustion (83%), sexual problems (74%), anxiety (72%), irritability (68%), depressive mood (67%), weight gain (67%), hot flushes (65%), and joint and muscular discomfort (63%). The symptoms that were perceived to most negatively affect training and performance were joint and muscular discomfort, weight gain, sleep problems, and physical and mental exhaustion. Participants with a perceived strong negative effect of symptoms on training and performance demonstrated higher total MRS scores compared to participants reporting no negative effect or slight negative effect of symptoms on training and performance (p's ≤ .001), suggesting women with more severe menopausal symptoms had a greater perceived negative effect of symptoms on training and performance. In summary, frequency of menopausal symptoms among female endurance athletes is high and severity of menopausal symptoms is similar to that reported in the general population. Despite high physical activity levels in this population, clinicians should address these symptoms to promote continued participation in physical activity.

**Data availability statement:** All relevant data are within the paper and its Supporting Information file.

**Funding:** This study was supported by the National Heart, Blood, Lung Institute in the form of a grant awarded to N.M.Y. (K01 HL159348-01) and Old Dominion University in the form of a salary for N.M.Y. The specific roles of this author are articulated in the 'author contributions' section. The funders had no role in study design, data collection and analysis, decision to publish, or preparation of the manuscript.

**Competing interests:** The authors have declared that no competing interests exist.

## Introduction

Menopause is an inevitable phase of life for the female population. The menopause transition is characterized by a steep decline in estrogen production, which is a significant alteration in the hormone profile compared to females in the reproductive phase [1]. Beginning in perimenopause, changes in follicle stimulating hormone and estradiol production contribute to variability in menstrual cycle length, and a female enters postmenopause after 12 months without menses [1]. As a result of the hormonal fluctuations occurring during perimenopause and into postmenopause, prevalence of menopausal symptoms is high during this time. Commonly reported symptoms include vasomotor symptoms, e.g., hot flushes, reported in up to 80% of females [2]; sleep problems, 40–60% [3]; depressive mood, 74.6% [4]; joint and muscular discomfort, 77% [4]; and genitourinary symptoms of menopause, e.g., vaginal dryness, urinary incontinence, 40–60% [5]. These symptoms have a negative effect on health-related quality of life and lead to higher healthcare utilization [6]. Considering increased life expectancy, this has significant population health and economic implications, with an estimated 73 million females either in menopausal transition or post-menopause in the United States [7]. However, it is unclear how specific menopausal symptoms may be affecting participation in physical activity.

Physical activity is an essential component of healthy aging, particularly for menopausal females [8]. Physical activity combats the musculoskeletal changes associated with menopause [9], reduces the risk of chronic disease [10], and improves quality of life in menopausal females [8]. Middle-aged females are encouraged to follow established recommendations for physical activity [11], including both aerobic exercise and resistance training for optimal health outcomes [12]. Females are increasingly participating in endurance sports (e.g., running, swimming, cycling) into older ages [13–16], but menopause has not been well-studied among female athletes [17]. While it is well-established that physical activity is beneficial in reducing menopausal symptoms [18,19], menopausal symptom prevalence is still high among physically active women [20]. Bothersome menopausal symptoms may still interfere with participation in physical activity and sport. In one of the few studies considering menopausal symptoms in female athletes, Ussher et al. (2008) studied female master swimmers and reported that perimenopausal swimmers had the greatest reduction in swimming volume and intensity compared to premenopausal and postmenopausal swimmers, with 24% of participants reporting that menopausal symptoms negatively affected their training [17]. What is unknown are specific barriers that are contributing to decreased volume and intensity of exercise in active menopausal females.

Because of the overwhelming benefits of physical activity in menopausal females [9,21–23], it is imperative to understand and address any potential physiological barriers to physical activity to provide targeted intervention strategies to promote participation in physical activity during this crucial time in mid-life. However, little is known about specific menopausal symptoms that menopausal athletes experience and how these symptoms may interfere with physical activity. Therefore, the purpose of this study was to determine the frequency and severity of menopausal symptoms

in female endurance athletes. Additionally, we examined the differences in menopausal symptom severity among participants who reported no negative effects, slight negative effects, moderate negative effects, and strong negative effects of menopausal symptoms on training and performance.

## Materials and methods

This study was reviewed and approved by the University Health Sciences Human Subjects Ethics Committee. Given that identifying information about participants was not collected, this study was determined to be exempt from requiring a signed informed consent document. However, a notification statement was provided, as requested and approved by the university ethics committee, which provided participants with a description of the study prior to data collection and notified that completion of the survey indicated informed consent to participate. Participants were recruited from February 6, 2024 through August 31, 2024 via social media, word of mouth, and local and regional athletic groups in the United States. Interested participants completed a brief screening survey via Qualtrics, and if participants met the inclusion criteria, an investigator emailed them the link to complete the survey. Inclusion criteria included (1) females ages 40–60 years, (2) participating in running, cycling, swimming, and/or triathlons for at least the past 5 years, (3) participating in running, cycling, and/or swimming at least 3 days per week, and (4) participating in a total of at least 3 hours of physical activity per week. Participants were excluded if they were male or if they did not meet minimum physical activity volume described in the inclusion criteria. The survey was administered anonymously through Qualtrics. Participants were required to complete the menopausal status and Menopausal Rating Scale questions for the questionnaire to be considered valid.

### Measures

**Menopausal status.** Participants self-reported their menopausal status as premenopausal ("continuing to experience regular menstrual cycles, or no changes in menstruation compared to what is usually experienced"), perimenopausal ("periods starting to become irregular; experiencing changes in frequency, length, or nature of bleeding in menstrual cycles"), or postmenopausal ("have not experienced a menstrual cycle in 12 months or more.") [1]. If participants reported being postmenopausal, they were asked if they had a natural or a surgical menopause. All participants were asked if they are currently taking hormone replacement therapy (HRT).

**Menopausal rating scale.** The Menopausal Rating Scale (MRS) has been validated for assessing menopausal symptoms and health-related quality of life in menopausal females [24]. Participants rated 11 menopausal symptoms (e.g., hot flushes, sleep problems, bladder problems) on a 5-point Likert scale, i.e., (0)-none, (1) mild, (2) moderate, (3) severe, or (4)-very severe [24,25]. A total menopausal symptom rating score was calculated by summing all 11 items. Based on results of a previous study investigating menopausal symptoms among Master swimmers [17], weight gain was also included as another menopausal symptom that was rated on the same 5-point Likert scale, but this symptom was not included in the total MRS score calculation.

**Perceived influence of menopausal symptoms on training and performance.** If a participant rated any MRS symptom as greater than "none," survey logic was used to ask the participant: "How do you feel that [symptom] has affected your *training* in your primary activity?" and "How do you feel that [symptom] has affected your *performance* in your primary activity?" Possible responses included: "[Symptom] has not affected my training/performance," "[Symptom] has a slight negative effect on my training/performance," "[Symptom] has a moderate negative effect on my training/performance," or "[Symptom] has a strong negative effect on my training/performance." All participants were asked about menopausal symptoms in general affecting training and performance with the questions: "In general, do you perceive that your menopausal symptoms have negatively affected your *training* in your primary activity?" and "In general, do you perceive that your menopausal symptoms have negatively affected your *performance* in your primary activity?" with possible responses including: "not experiencing menopausal symptoms," "no negative effect of menopausal symptoms on training/performance," "slight negative effect of menopausal symptoms on training/performance," "moderate negative

effect of menopausal symptoms on training/performance," and "strong negative effect of menopausal symptoms on training/performance." Finally, participants were asked about their general health: "In general, would you say your health is:" "excellent," "very good," "good," "fair," or "poor."

*Statistical analysis*. Proportions and descriptive statistics were primarily used to report the data. Differences in age and total MRS score among pre-, peri-, and postmenopausal groups were assessed using Kruskal-Wallis test, as data were not normally distributed. Kruskal-Wallis was also used to determine differences in total MRS scores among participants reporting no negative effect, slight negative effect, moderate negative effect, and strong negative effect of menopausal symptoms on training and performance. IBM SPSS Statistics (version 29) was used for all statistical analyses and level of significance was set at $p < .05$.

## Results

Out of 384 completed screening surveys received, 265 potential participants met inclusion criteria and were emailed the link to complete the survey. There were 187 female endurance athletes who participated in this survey study, yielding a 71% response rate. The most common primary endurance sport among all participants was running (55%, n = 102), followed by swimming (22%, n = 41), cycling (14%, n = 27), and triathlon (9%, n = 17). Eighteen percent (n = 34) reported being premenopausal, 39% (n = 73) reported being perimenopausal, and 43% (n = 80) reported being postmenopausal. Fifty-three participants (28%) reported current HRT use; most of these were perimenopausal (n = 22) or postmenopausal (n = 29). Most of the postmenopausal participants (n = 66, 83%) reported a natural menopause.

Frequency of menopausal symptoms among our sample of female endurance athletes is high (Table 1). The most common menopausal symptoms reported were sleep problems (88%, n = 164) and physical and mental exhaustion (83%, n = 155). Other common symptoms reported were sexual problems (74%, n = 139), anxiety (72%, n = 136), irritability (68%, n = 128), depressive mood (67%, n = 125), weight gain (67%, n = 126), hot flushes (65%, n = 122), and joint and muscular discomfort (63%, n = 118). Other reported symptoms include vaginal dryness (59%, n = 110), bladder problems (52%, n = 97), and heart discomfort (41%, n = 77). Most participants reported these symptoms as mild or moderate, with few participants rating the symptoms as severe or very severe (Table 1).

Several menopausal symptoms were reported to negatively affect training (Table 2) and performance (Table 3). The symptoms most likely to be perceived to have a negative effect on training and performance are joint and muscular discomfort (97% reporting a negative effect on training and 91% reporting a negative effect on performance), sleep problems (92% reporting a negative effect on training and 89% reporting a negative effect on performance), physical and mental exhaustion (87% reporting a negative effect on training and 88% reporting a negative effect on performance), and weight gain (79% reporting a negative effect on training and 88% reporting a negative effect on performance). In response to the questions, "In general, do you perceive that your menopausal symptoms have negatively affected your *training/performance* in your primary activity?", more perimenopausal and postmenopausal participants reported a general negative effect of menopausal symptoms on training (Fig 1) and performance (Fig 2) compared to premenopausal participants. The vast majority of all participants reported their general health as "excellent" (38%, n = 70) or "very good" (46%, n = 84) (Fig 3).

There was a significant effect of menopausal group on age ($p < .001$, Table 4), with postmenopausal participants significantly older than premenopausal participants ($p < .001$) and perimenopausal participants ($p < .001$), and perimenopausal participants significantly older than premenopausal participants ($p = .011$). There was also a significant effect of menopausal group on total MRS score ($p < .001$, Table 4). Pairwise comparisons show that total MRS score is significantly higher in perimenopausal females compared to premenopausal females ($p < .001$) and postmenopausal participants ($p = .009$), and total MRS score was greater for postmenopausal participants compared to premenopausal participants ($p = .005$). Total MRS scores were significantly different across participants reporting no negative effect, slight negative

**Table 1. Menopausal Rating Scale scores among pre-, peri-, and postmenopausal female endurance athletes.**

| Menopausal symptom | None | | | Mild | | | Moderate | | | Severe | | | Very Severe | | | Total reporting symptom (n = 187) |
|---|---|---|---|---|---|---|---|---|---|---|---|---|---|---|---|---|
| | Pre | Peri | Post | Pre | Peri | Post | Pre | Peri | Post | Pre | Peri | Post | Pre | Peri | Post | |
| Hot flushes | 62% (21) | 22% (16) | 35% (28) | 32% (11) | 52% (38) | 41% (33) | 6% (2) | 22% (16) | 20% (16) | 0% (0) | 4% (3) | 3% (2) | 0% (0) | 0% (0) | 1% (1) | 65% (122) |
| Dryness of vagina | 56% (19) | 47% (34) | 30% (24) | 29% (10) | 34% (25) | 23% (18) | 15% (5) | 11% (8) | 30% (24) | 0% (0) | 6% (4) | 11% (9) | 0% (0) | 3% (2) | 6% (5) | 59% (110) |
| Joint and muscular discomfort | 56% (19) | 30% (22) | 35% (28) | 35% (12) | 30% (22) | 34% (27) | 9% (3) | 27% (20) | 25% (20) | 0% (0) | 11% (8) | 4% (3) | 0% (0) | 1% (1) | 3% (2) | 63% (118) |
| Heart discomfort | 68% (23) | 48% (35) | 65% (52) | 29% (10) | 36% (26) | 21% (17) | 3% (1) | 14% (10) | 13% (10) | 0% (0) | 1% (1) | 1% (1) | 0% (0) | 1% (1) | 0% (0) | 41% (77) |
| Sleep problems | 24% (8) | 7% (5) | 13% (10) | 32% (11) | 33% (24) | 29% (23) | 35% (12) | 37% (27) | 43% (34) | 9% (3) | 19% (14) | 16% (13) | 0% (0) | 4% (3) | 0% (0) | 88% (164) |
| Depressive mood | 35% (12) | 19% (14) | 45% (36) | 53% (18) | 38% (28) | 33% (26) | 12% (4) | 34% (25) | 16% (13) | 0% (0) | 8% (6) | 6% (5) | 0% (0) | 0% (0) | 0% (0) | 67% (125) |
| Irritability | 41% (14) | 15% (11) | 43% (34) | 35% (12) | 38% (28) | 41% (33) | 24% (8) | 37% (27) | 13% (10) | 0% (0) | 8% (6) | 3% (2) | 0% (0) | 1% (1) | 1% (1) | 68% (128) |
| Anxiety | 29% (10) | 15% (11) | 38% (30) | 53% (18) | 37% (27) | 38% (30) | 15% (5) | 41% (30) | 20% (16) | 3% (1) | 6% (4) | 5% (4) | 0% (0) | 1% (1) | 0% (0) | 72% (136) |
| Physical and mental exhaustion | 29% (10) | 7% (5) | 21% (17) | 29% (10) | 33% (24) | 48% (38) | 38% (13) | 41% (30) | 21% (17) | 3% (1) | 12% (9) | 9% (7) | 0% (0) | 7% (5) | 1% (1) | 83% (155) |
| Sexual problems | 38% (13) | 16% (12) | 29% (23) | 38% (13) | 41% (30) | 20% (16) | 21% (7) | 22% (16) | 28% (22) | 3% (1) | 15% (11) | 14% (11) | 0% (0) | 6% (4) | 10% (8) | 74% (139) |
| Bladder problems | 74% (25) | 43% (31) | 43% (34) | 23% (8) | 26% (19) | 34% (27) | 3% (1) | 23% (17) | 18% (14) | 0% (0) | 6% (4) | 5% (4) | 0% (0) | 3% (2) | 1% (1) | 52% (97) |
| Weight gain | 56% (19) | 29% (21) | 26% (21) | 29% (10) | 36% (26) | 34% (27) | 9% (3) | 20% (15) | 25% (20) | 6% (2) | 8% (6) | 11% (9) | 0% (0) | 7% (5) | 4% (3) | 67% (126) |

Scores reported as: total percentage of premenopausal (Pre; n = 34), perimenopausal (Peri; n = 73), and postmenopausal (Post; n = 80) participants reporting score (n).

effect, moderate negative effect, and strong negative effect of menopausal symptoms on training ($p < .001$) and performance ($p < .001$), suggesting that a higher reported severity of menopausal symptoms has a greater perceived negative effect on training and performance (Table 5). Pairwise comparisons show that total MRS score is significantly higher among participants reporting a strong negative effect of menopausal symptoms on training compared to those reporting a slight negative effect ($p = .001$) and those reporting no negative effect of symptoms on training ($p < .001$), and higher among participants reporting a moderate negative effect of symptoms on training compared to those reporting a slight negative effect ($p = .007$) and no negative effect of symptoms on training ($p < .001$). Pairwise comparisons also show that total MRS score is significantly higher among participants reporting a strong negative effect of symptoms on performance compared to those reporting a slight negative effect ($p = .001$) and those reporting no negative effect of symptoms on performance ($p < .001$), and higher among participants reporting a moderate negative effect of symptoms on performance compared to those reporting no negative effect of symptoms on performance ($p = .001$).

## Discussion

This study examined the frequency of menopausal symptoms among female endurance athletes and found that although most of the participants reported their overall health as excellent to very good, menopausal symptoms were still prevalent in this population. The most common menopausal symptoms reported among participants were sleep problems and

**Table 2. Perceived effect of menopausal symptoms on training in female endurance athletes.**

| Menopausal symptom | No effect | Slight negative effect | Moderate negative effect | Strong negative effect |
|---|---|---|---|---|
| Hot flushes (n = 118) | 67% (79) | 28% (33) | 5% (6) | 0% (0) |
| Heart discomfort (n = 75) | 55% (41) | 33% (25) | 8% (6) | 4% (3) |
| Joint and muscular discomfort (n = 116) | 3% (3) | 39% (45) | 41% (47) | 18% (21) |
| Sleep problems (n = 160) | 8% (12) | 46% (73) | 38% (60) | 9% (15) |
| Physical and mental exhaustion (n = 153) | 13% (20) | 41% (63) | 34% (52) | 12% (18) |
| Depressive mood (n = 122) | 32% (39) | 44% (54) | 15% (18) | 9% (11) |
| Irritability (n = 124) | 51% (63) | 36% (44) | 10% (13) | 3% (4) |
| Anxiety (n = 132) | 40% (53) | 43% (57) | 13% (17) | 4% (5) |
| Bladder problems (n = 95) | 46% (44) | 36% (34) | 13% (12) | 5% (5) |
| Sexual problems (n = 137) | 93% (128) | 7% (9) | 0% (0) | 0% (0) |
| Dryness of vagina (n = 108) | 86% (93) | 11% (12) | 2% (2) | 1% (1) |
| Weight gain (n = 124) | 21% (26) | 35% (44) | 29% (36) | 15% (18) |

Results reported as: total percentage of participants reporting score (n).

physical and mental exhaustion. The menopausal symptoms that had the greatest perceived negative effect on training and performance were joint and muscular discomfort, sleep problems, and physical and mental exhaustion. Participants reporting a greater perceived negative effect of symptoms on training and performance demonstrated greater severity of menopausal symptoms compared to those reporting no or lesser negative effects of menopausal symptoms on training and performance.

Our findings regarding rates of menopausal symptoms are generally consistent with those from general samples of females. Sleep disturbances and fatigue are common among menopausal females, affecting 40–72% of the menopausal female population [4,26,27]. In menopausal females, poor sleep can negatively affect cardiovascular health [28], mood [29,30], and quality of life [31]. Poor sleep is also likely interfering with participation in physical activity, as 49% of women from a general sample aged 40–65 years report "feeling too tired" as a barrier to participation in physical activity [32]. In two of the few studies investigating menopause among female athletes, Ussher et al. (2008) reported sleeplessness as one of the most frequently reported symptoms among menopausal master swimmers (47%) [17], and in a study including menopausal weightlifting athletes, Huebner et al. (2020) reported that the two most common symptoms were trouble sleeping (43%) and fatigue (34%) [33]. The very high frequency of sleep problems (88%) and exhaustion (83%) in our sample is particularly concerning because of the importance of sleep for both sport performance and prevention of injury in athletes [34–36]. Although exercise is known to improve sleep in menopausal females [37], our results highlight the potential utility of interventions targeting improved sleep for menopausal female athletes who are already regularly participating in physical activity. Research supports cognitive behavioral therapy, mindfulness and relaxation, yoga, pharmacologic intervention, and nutritional intervention for improved sleep among menopausal

**Table 3. Perceived effect of menopausal symptoms on performance in female endurance athletes.**

| Menopausal symptom | No effect | Slight negative effect | Moderate negative effect | Strong negative effect |
|---|---|---|---|---|
| Hot flushes (n=118) | 72% (85) | 20% (24) | 8% (9) | 0% (0) |
| Heart discomfort (n=75) | 57% (43) | 29% (22) | 9% (7) | 4% (3) |
| Joint and muscular discomfort (n=116) | 9% (10) | 37% (43) | 35% (41) | 19% (22) |
| Sleep problems (n=160) | 11% (17) | 48% (76) | 33% (52) | 9% (15) |
| Physical and mental exhaustion (n=153) | 12% (18) | 42% (65) | 33% (51) | 12% (19) |
| Depressive mood (n=122) | 35% (43) | 44% (54) | 12% (15) | 8% (10) |
| Irritability (n=124) | 55% (68) | 31% (39) | 10% (13) | 3% (4) |
| Anxiety (n=132) | 43% (57) | 38% (50) | 12% (16) | 7% (9) |
| Bladder problems (n=95) | 56% (53) | 31% (29) | 8% (8) | 5% (5) |
| Sexual problems (n=137) | 93% (128) | 6% (8) | 1% (1) | 0% (0) |
| Dryness of vagina (n=108) | 89% (96) | 9% (10) | 2% (2) | 0% (0) |
| Weight gain (n=124) | 12% (15) | 40% (50) | 30% (37) | 18% (22) |

Results reported as: total percentage of participants reporting score (n).

females [29,38–40], and sleep extension and sleep hygiene are recommended for improved sleep among athletes [41–43]; however, these strategies require further investigation to assess their effectiveness specifically for female menopausal athletes.

In addition to sleep and physical and mental exhaustion, joint and muscular discomfort and weight gain had the strongest perceived negative effect on training and performance. One systematic review concluded that athletes with musculoskeletal injuries and subsequent pain had high levels of depressive symptoms, though most of the reviewed studies included younger athletes (ages 18–27 years) and only 21% of included participants were female [44]. The association between musculoskeletal pain and depression symptoms is notable because of the high frequency of self-reported depressive mood in our sample (67%) in response to the MRS. These musculoskeletal symptoms may have a compounding effect with the psychological symptoms, ultimately negatively affecting training and performance in menopausal athletes. Wright et al. (2024) recently introduced the terminology "musculoskeletal syndrome of menopause" to describe the constellation of factors affecting joint and muscle health during menopause (i.e., inflammation, decreased muscle mass and bone mineral density, and arthritis) [45]. Considering that approximately 71% of perimenopausal women have musculoskeletal pain [46], and in our study of athletes we found 63% reported musculoskeletal pain, it is imperative for rehabilitation clinicians and healthcare providers to be aware of this syndrome in order to effectively address this common condition among menopausal females.

Perimenopause is characterized by high variability in hormonal fluctuations compared to females in the reproductive phase [1]. Unsurprisingly, perimenopausal participants reported more severe menopausal symptoms compared

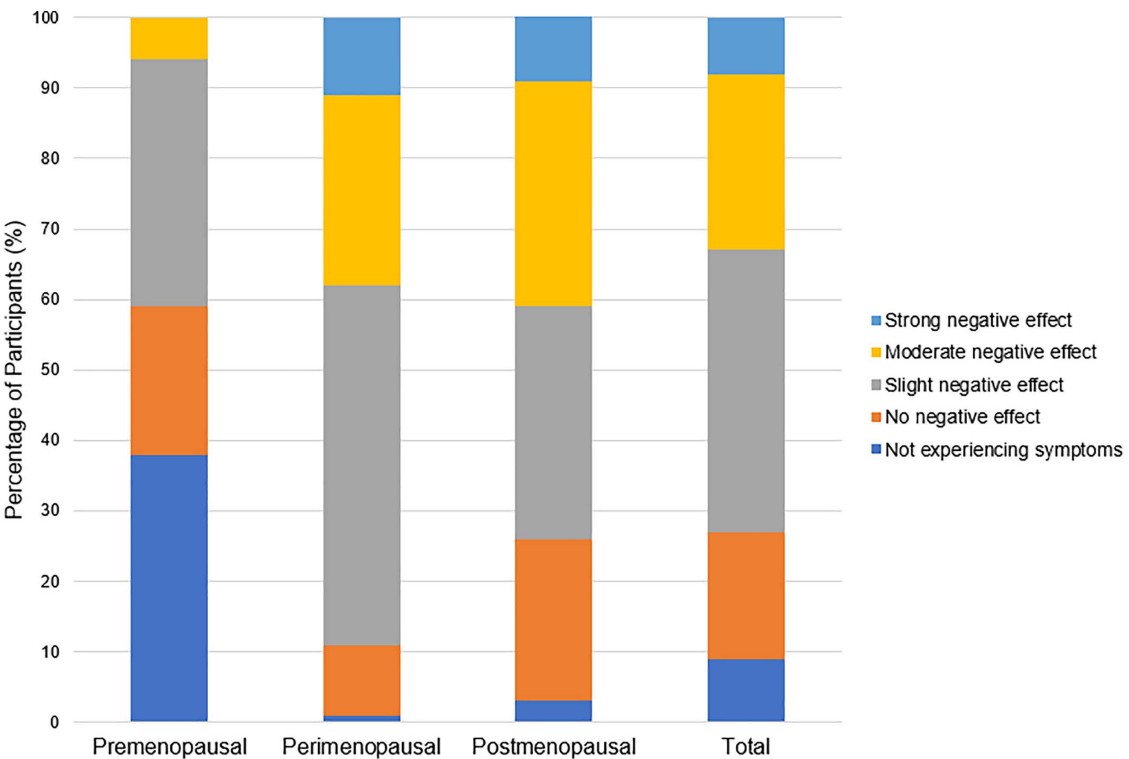

**Fig 1. Perceived influence of menopausal symptoms on training in pre-, peri-, and postmenopausal participants (n = 183).**

to pre- and postmenopausal participants. Similarly, more peri- and postmenopausal participants reported that their menopausal symptoms had a moderate or strong negative effect on their training and performance compared to pre-menopausal participants, and across the entire sample, greater reported severity of menopausal symptoms predicted a greater perceived negative effect on training and performance. Total MRS scores in the peri- (14.2 ± 6.0) and post-menopausal (11.7 ± 6.1) participants are considered "moderate" severity of symptoms (total score range 9–15) [24]. Despite evidence in support of exercise for improving menopausal symptoms in the general female population [47–50], reported MRS scores in our sample of highly physically active participants are similar to a range of total MRS scores reported in the general population [4,24,51]. Because of the positive benefits of physical activity in improving cardiovascular health [52], musculoskeletal health [9], and quality of life [8] in menopausal females, it is imperative to educate this population about the importance of continuing a practice of regular exercise. Unfortunately, in a survey of Strava app (exercise activity tracking application) users, only 11% of participants reported receiving advice regarding exercise during menopause [20]. Perimenopause is a crucial time for athlete education regarding common menopausal symptoms and interventions to address these symptoms in order to facilitate continued participation in physical activity.

Strengths of this study include minimum physical activity volume of participants to ensure a physically active sample. We also used a validated measure to assess and report a range of menopausal symptoms in our participants, allowing comparative analysis to other menopausal samples. Finally, this is one of the first studies to investigate menopausal symptoms among highly active menopausal females. Study limitations include the cross-sectional nature of this data collection, which prevents drawing conclusions about the causal associations between menopausal symptoms, training,

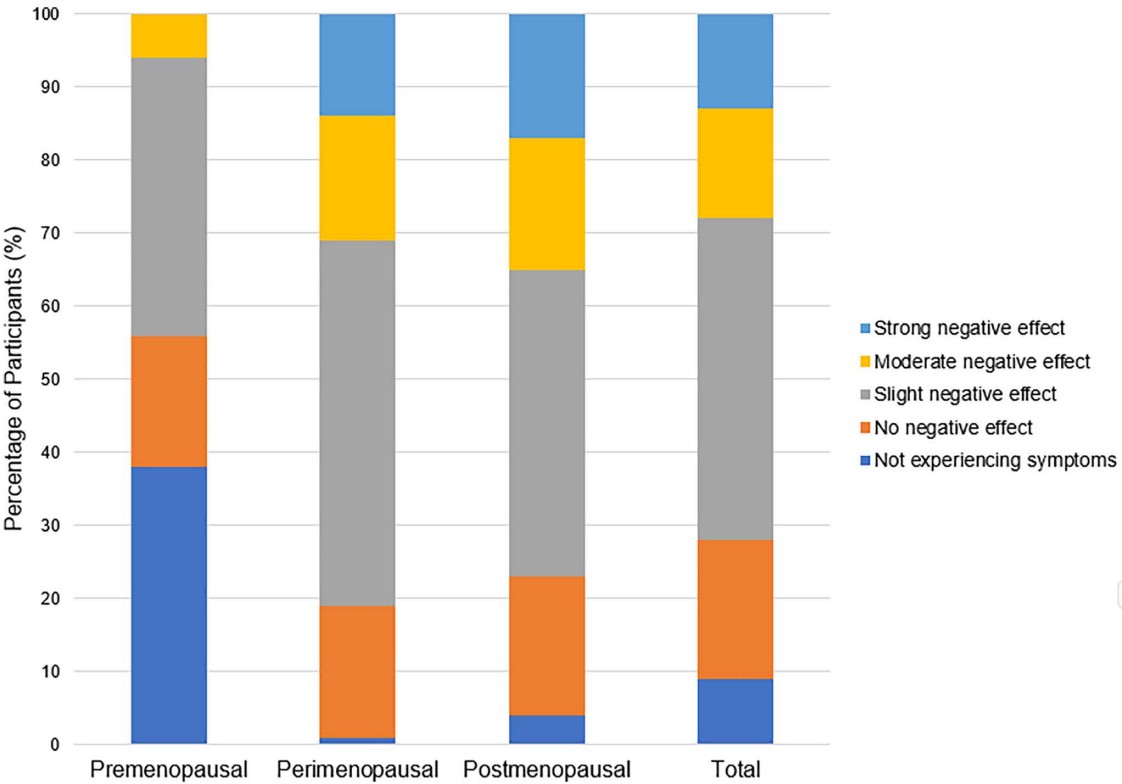

**Fig 2. Perceived influence of menopausal symptoms on performance in pre-, peri-, and postmenopausal participants (n = 183).**

and performance. Additionally, we did not collect demographic information (e.g., race/ethnicity, level of education, income), which makes it difficult to interpret the generalizability of these results to the broader population. Although menopausal status was self-reported, we provided menopausal status definitions that are consistent with the Stages of Reproductive Aging Workshop criteria [1]. Finally, there may have been other personal factors that are common in middle-aged females (e.g., work-related stress, caregiving responsibilities, other health conditions) contributing to the symptoms reported in our sample. We did not collect health history, so there may be underlying medical or psychological conditions contributing to these symptoms. The use of hormone replacement therapy, medications, and/or supplements can affect menopausal symptoms and may have influenced results. However, regardless of the multifaceted cause of these symptoms, the high frequency of reported symptoms should be addressed in this population. Future studies with strong research design and larger sample sizes are needed to confirm our findings and to determine the most effective interventions to address these symptoms.

In conclusion, despite evidence suggesting that exercise improves menopausal symptoms, frequency of menopausal symptoms among female endurance athletes is high and severity of menopausal symptoms is similar to that reported in the general population. Sleep problems and physical and mental exhaustion were the most common symptoms reported, and together with joint and muscular discomfort and weight gain had the greatest perceived negative effect on training and performance. Larger sample sizes with high quality research design are needed to confirm these findings. Healthcare providers and rehabilitation clinicians need to address these symptoms in this population to promote continued participation in physical activity.

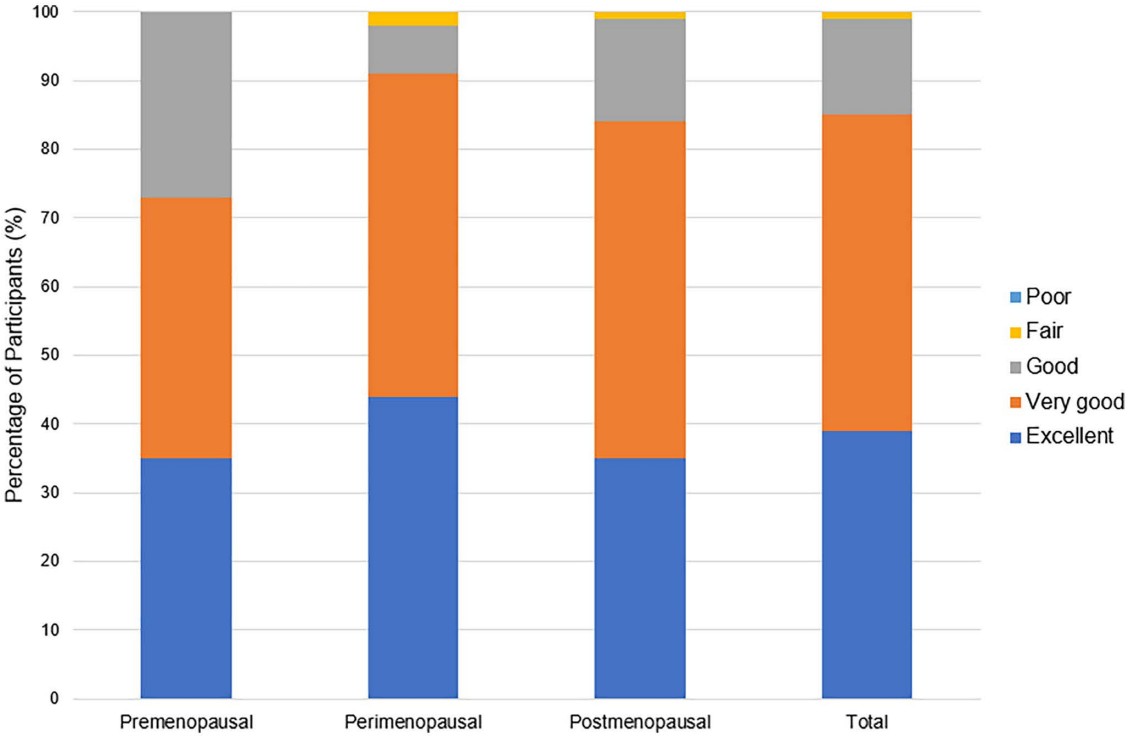

**Fig 3. Reported general health in pre-, peri-, and postmenopausal participants (n = 182).**

**Table 4. Age and total Menopausal Rating Scale scores across participants.**

| Participant Characteristics | Premenopausal (n = 34) | Perimenopausal (n = 73) | Postmenopausal (n = 80) | All participants (n = 187) | p |
|---|---|---|---|---|---|
| Age (years) | 44.4 ± 3.2 | 47.4 ± 3.7 | 55.2 ± 4.2 | 50.2 ± 5.9 | <.001* |
| Total MRS score | 8.0 ± 4.7 | 14.2 ± 6.0 | 11.7 ± 6.1 | 12.0 ± 6.2 | <.001* |

M ± SD = mean ± standard deviation.

*$p < .05$.

**Table 5. Total Menopausal Rating Scale scores across perceived effect of menopausal symptoms on training and performance.**

| | No negative effect | Slight negative effect | Moderate negative effect | Strong negative effect | p |
|---|---|---|---|---|---|
| Perceived effect of menopausal symptoms on training | 8.5 ± 4.7 | 11.4 ± 4.3 | 15.1 ± 5.5 | 19.5 ± 7.8 | <.001* |
| Perceived effect of menopausal symptoms on performance | 8.8 ± 5.0 | 12.1 ± 4.7 | 14.1 ± 5.7 | 18.4 ± 7.0 | <.001* |

M ± SD = mean ± standard deviation

*$p < .05$

# Supporting information

### S1 File. Female endurance athlete survey data.
(XLSX)

## Author contributions

**Conceptualization:** Heather M. Hamilton, Natalie M. Yarish.

**Formal analysis:** Heather M. Hamilton, Natalie M. Yarish.

**Investigation:** Heather M. Hamilton, Natalie M. Yarish.

**Methodology:** Heather M. Hamilton, Natalie M. Yarish.

**Project administration:** Heather M. Hamilton.

**Supervision:** Heather M. Hamilton.

**Writing – original draft:** Heather M. Hamilton, Natalie M. Yarish.

**Writing – review & editing:** Heather M. Hamilton, Natalie M. Yarish, Kristin E. Heron.

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
