## [Decision Letter · Decision Letter 0]

1 Jul 2025

Dear Dr. Hamilton,

Thank you for submitting your manuscript to PLOS ONE. After careful consideration, we feel that it has merit but does not fully meet PLOS ONE’s publication criteria as it currently stands. Therefore, we invite you to submit a revised version of the manuscript that addresses the points raised during the review process.

We look forward to receiving your revised manuscript.

Kind regards,

Luiz F. Baccaro

Academic Editor

PLOS ONE

Journal Requirements: 

2. We note that your Data Availability Statement is currently as follows: [Data is not publicly available because participants were not made aware in advance of the potential for individual data to be published. Interested researchers are welcome to contact the corresponding author, and reasonable requests for data will be considered on a case-by-case basis.]

Reviewers' comments:

Reviewer's Responses to Questions

**Comments to the Author**

1. Is the manuscript technically sound, and do the data support the conclusions?

Reviewer #1: No

Reviewer #2: Yes

Reviewer #3: No

2. Has the statistical analysis been performed appropriately and rigorously?

Reviewer #1: No

Reviewer #2: Yes

Reviewer #3: Yes

3. Have the authors made all data underlying the findings in their manuscript fully available?

Reviewer #1: No

Reviewer #2: Yes

Reviewer #3: No

4. Is the manuscript presented in an intelligible fashion and written in standard English?

Reviewer #1: Yes

Reviewer #2: Yes

Reviewer #3: Yes

Reviewer #1: The study was based on a survey study with designed question items.

It is inappropriate to use linear regression to study the association between total MRS score and perceived effect of symptoms. The outcome in here should not be regarded as a continuous variable with normal density as is usually assumed in linear regression models.

I don’t think the information generated from this study is very useful. Most results are based on summary statistics calculated for different menopausal symptoms. For example, you established the significance of age variable. But that is a well-known result documented in the medical literature.

The sample size is moderate.

Reviewer #2: The authors present a very interesting cross-sectional study with the aim of determining the prevalence of menopausal symptoms in female endurance athletes and examining the perceived influence of menopausal symptoms on the training and performance of this population. Some suggestions are made below.

In order to talk about the prevalence of menopausal symptoms in the study population, we need to know if the sample is representative of the total universe of these women in the United States. In the methodology, the authors need to provide an approximate number of American women who practice the physical activities considered in the methodology. Also, how did they arrive at the figure of 384 screening surveys presented in the results section? Is this the approximate number of American women who do these physical activities? Why didn't the authors ask if the research participants use any medication that improves menopause-related symptoms? Why was this not an exclusion criterion in this study?

In the results section, as the sample of women who responded to the survey does not seem to be representative of the female population who practice the physical activities considered in this study, it would be more correct to change the term prevalence to frequency, as this is a descriptive study and not a prevalence study.

There is no need to repeat the study's proposal at the beginning of the discussion, but I would urge the authors to rethink changing the term prevalence to frequency in the introduction and results. I suggest that the first paragraph of the discussion be limited to the text: “The most common menopausal symptoms reported among participants were sleep problems and physical and mental exhaustion. The menopausal symptoms that had the greatest perceived negative effect on training and performance were joint and muscular discomfort, sleep problems, and physical and mental exhaustion. Greater reported severity of menopausal symptoms was associated with a greater perceived negative effect of symptoms on training and performance.” Finally, there may have been other personal factors that are common in middle-aged females (e.g., work-related stress, caregiving responsibilities, other health conditions) contributing to the symptoms reported in our sample. We did not collect health history, so there may be underlying medical or psychological conditions contributing to these symptoms.”, the authors should add that if there was inclusion of women who use hormone replacement, antidepressants, herbal medicines and other supplements that can influence menopausal symptoms, this bias may have influenced the results.

Finally, the conclusion paragraph should state that the frequency of menopausal symptoms in the population that responded to the email was high, objectively complementing which symptoms were associated with a negative effect on training and performance. It should also include the prospect of new studies with a better design so that the results of this study can be confirmed with a better level of evidence.

Reviewer #3: The study rationale must also be better justified. Research that investigates the impact of physical activity during menopause in greater depth is both urgent and highly relevant, especially considering the increasing life expectancy of women, growing concern with quality of life, and global population aging.

Reviewer’s CommentThe article addresses a highly relevant and globally significant topic, with recent scientific evidence highlighting physical activity as a key component in promoting health among menopausal women. The manuscript focuses on the prevalence of menopause and the most commonly reported symptoms experienced during this phase, contributing to the understanding of the clinical and psychosocial impact of the menopausal transition. However, the introduction is unclear and lacks coherence. It fails to adequately contextualize the topic or provide foundational background, such as citing established guidelines from the North American Menopause Society (NAMS) and the International Federation of Gynecology and Obstetrics (FIGO). These guidelines emphasize the importance of regular aerobic and resistance physical activity in alleviating menopausal symptoms, improving quality of life, and reducing morbidity and mortality associated with common comorbidities. Moreover, the introduction should include references that emphasize the relevance of the topic and identify existing gaps in the literature. For example, key conceptual definitions are only introduced later in the discussion section, which weakens the logical structure of the manuscript. The study rationale must also be better justified. Research that investigates the impact of physical activity during menopause in greater depth is both urgent and highly relevant, especially considering the increasing life expectancy of women, growing concern with quality of life, and global population aging.Materials and MethodsSeveral methodological points need clarification:- Exclusion criteria are not specified.- Was surgical menopause considered in participant selection?- Regarding the Menopause Rating Scale (MRS), it is recommended that the full 5-point Likert scale used in the study be described.- The handling of missing data is unclear. It should be stated whether participants were required to complete all questions for the questionnaire to be considered valid.ResultsThis section requires several improvements for clearer and more effective presentation of the data:- The statement “Prevalence of menopausal symptoms among female endurance athletes is high” must be clarified or rewritten to reflect the data accurately.- Figures and tables present overlapping information. Therefore:  - Figure 1 and Table 1 convey the same content; I recommend removing Figure 1.  - Table 1 should present results stratified by menopausal status.  - Figures 2, 3, 4, and 5 should be removed, as their data are already included in Table 2.  - Table 2 requires improvement to enhance clarity and interpretability.  - In the paragraph stating “Several menopausal symptoms were reported to negatively affect training”, I recommend summarizing only the key findings, as the detailed information is already well presented in the accompanying table. Discussion The discussion is overly long and lacks comparative analysis with findings from other studies involving menopausal women. While the authors provide a detailed account of the study's limitations, the manuscript would benefit from a more balanced perspective by also highlighting the strengths of the study. This would enhance the critical value and credibility of the discussion and contribute to the interpretation of the findings in a broader scientific context.

**Do you want your identity to be public for this peer review?** For information about this choice, including consent withdrawal, please see our Privacy Policy

Reviewer #1: No

Reviewer #2: **Yes: ** Ricardo Ney Cobucci

Reviewer #3: No

---

## [Author Response · Author response to Decision Letter 1]

12 Sep 2025

Author Responses to Reviewer Comments

The authors thank the reviewers for their thoughtful review and helpful suggestions. In the document below we have included the comments individually followed by the authors’ responses to the comments. Requested changes to the manuscript have been saved as track changes, and line numbers in our responses below refer to the track changes version of the manuscript.

Reviewer #1:

1. The study was based on a survey study with designed question items.

It is inappropriate to use linear regression to study the association between total MRS score and perceived effect of symptoms. The outcome in here should not be regarded as a continuous variable with normal density as is usually assumed in linear regression models.

Author response: Thank you for this suggestion. We removed the linear regression analysis and instead used Kruskal-Wallis tests to evaluate differences in total MRS scores across groups reporting effects of menopausal symptoms on training and performance (no negative effect, slight negative effect, moderate negative effect, and strong negative effect of symptoms on training/performance). This analysis was included in the abstract, methods, results, and discussion sections, and a table (Table 5) was added to demonstrate mean and standard deviations of total MRS scores across these groups. The overall interpretation of these results is similar, in that participants with a greater perceived negative effect of symptoms demonstrate greater severity of menopausal symptoms compared to those reporting no / slight negative effect of menopausal symptoms on training and performance.

2. I don’t think the information generated from this study is very useful. Most results are based on summary statistics calculated for different menopausal symptoms. For example, you established the significance of age variable. But that is a well-known result documented in the medical literature.

Author response: We agree that this study replicated some established findings in the literature. However, very few studies have investigated menopausal symptoms among an athletic population, and this was the primary purpose of our study. The high frequency of menopausal symptoms in our sample draws attention to the need to screen for and address menopausal symptoms among highly active women to support continued participation in physical activity. To address this comment, in the revised manuscript we have provided additional information about what our study adds to the existing literature at the end of the introduction section.

3. The sample size is moderate.

Author response: We agree with the reviewer that the sample size is moderate. However, as noted above, we believe this study adds to the very small literature on menopausal female athletes and thus, contributes to this growing field. In addition, we have added a note in the limitations section that future research with larger samples is needed.

Reviewer #2:

1. The authors present a very interesting cross-sectional study with the aim of determining the prevalence of menopausal symptoms in female endurance athletes and examining the perceived influence of menopausal symptoms on the training and performance of this population. Some suggestions are made below.

Author response: Thank you for this positive feedback on our manuscript and for the suggestions on ways to improve. We have addressed each below.

2. In order to talk about the prevalence of menopausal symptoms in the study population, we need to know if the sample is representative of the total universe of these women in the United States. In the methodology, the authors need to provide an approximate number of American women who practice the physical activities considered in the methodology. Also, how did they arrive at the figure of 384 screening surveys presented in the results section? Is this the approximate number of American women who do these physical activities? Why didn't the authors ask if the research participants use any medication that improves menopause-related symptoms?

Why was this not an exclusion criterion in this study?

Author response: We have changed “prevalence” to “frequency” throughout the manuscript (including the title) for clarity.

There were 384 screening surveys that were received and reviewed by the investigators. This was clarified in the Results: “Out of 384 completed screening surveys received…” (line 162).

While we did not exclude participants for using hormone therapy or any specific medications, we did ask participants whether they use hormone therapy or not. These details were added to the results: “Fifty-three participants (28%) reported current HRT use; most of these were perimenopausal (n = 22) or postmenopausal (n = 29)” (lines 168-169). We wanted these results to be generalizable to a heterogeneous menopausal population, regardless of medication use.

3. In the results section, as the sample of women who responded to the survey does not seem to be representative of the female population who practice the physical activities considered in this study, it would be more correct to change the term prevalence to frequency, as this is a descriptive study and not a prevalence study.

Author response: We have changed “prevalence” to “frequency” throughout the manuscript (including the title) for clarity.

4. There is no need to repeat the study's proposal at the beginning of the discussion, but I would urge the authors to rethink changing the term prevalence to frequency in the introduction and results. I suggest that the first paragraph of the discussion be limited to the text: “The most common menopausal symptoms reported among participants were sleep problems and physical and mental exhaustion. The menopausal symptoms that had the greatest perceived negative effect on training and performance were joint and muscular discomfort, sleep problems, and physical and mental exhaustion. Greater reported severity of menopausal symptoms was associated with a greater perceived negative effect of symptoms on training and performance.”

Author response: Thank you for this feedback. We have revised the first sentence of the discussion to remove repeating all of the study aims. This sentence now reads: “This study examined the frequency of menopausal symptoms among female endurance athletes and found that although most of the participants reported their overall health as excellent to very good, menopausal symptoms were still prevalent in this population” (lines 274-277).

5. “Finally, there may have been other personal factors that are common in middle-aged females (e.g., work-related stress, caregiving responsibilities, other health conditions) contributing to the symptoms reported in our sample. We did not collect health history, so there may be underlying medical or psychological conditions contributing to these symptoms.”, the authors should add that if there was inclusion of women who use hormone replacement, antidepressants, herbal medicines and other supplements that can influence menopausal symptoms, this bias may have influenced the results.

Author response: We appreciate this point, and as noted above in response to this reviewer, we did not exclude women who were using hormone therapy or medications to reduce menopausal symptoms in order to have a more representative sample of menopausal women. However, the point is well-taken that it is plausible that some women’s symptoms may have been affected by such medications. To address this concern, in addition to adding data regarding the number of women using these medications (see response above), a statement was added to the limitations about the use of interventions that could influence results: “The use of hormone replacement therapy, medications, and/or supplements can affect menopausal symptoms and may have influenced the results” (Lines 356-357).

6. Finally, the conclusion paragraph should state that the frequency of menopausal symptoms in the population that responded to the email was high, objectively complementing which symptoms were associated with a negative effect on training and performance. It should also include the prospect of new studies with a better design so that the results of this study can be confirmed with a better level of evidence.

Author response: The conclusion was rephrased to highlight that the most common symptoms were also the ones with the greatest perceived negative effect on training/performance: “Sleep problems and physical and mental exhaustion were the most common symptoms reported, and together with joint and muscular discomfort and weight gain had the greatest perceived negative effect on training and performance” (lines 364-368). A statement was added to the conclusion paragraph about needing larger sample sizes and high quality research design to confirm these findings (lines 368-369).

Reviewer #3:

1. The study rationale must also be better justified. Research that investigates the impact of physical activity during menopause in greater depth is both urgent and highly relevant, especially considering the increasing life expectancy of women, growing concern with quality of life, and global population aging.

Author response: We agree with the reviewer that research examining the impact of activity during menopause is an important topic to study. The introduction section was revised to provide greater depth of background information and current knowledge gaps to support our study rationale. In particular, we now say: “Considering increased life expectancy, this has significant population health and economic implications, with an estimated 73 million females either in menopausal transition or post-menopause in the United States” (lines 64-66).

Reviewer’s Comments

1. The article addresses a highly relevant and globally significant topic, with recent scientific evidence highlighting physical activity as a key component in promoting health among menopausal women.

The manuscript focuses on the prevalence of menopause and the most commonly reported symptoms experienced during this phase, contributing to the understanding of the clinical and psychosocial impact of the menopausal transition. However, the introduction is unclear and lacks coherence. It fails to adequately contextualize the topic or provide foundational background, such as citing established guidelines from the North American Menopause Society (NAMS) and the International Federation of Gynecology and Obstetrics (FIGO). These guidelines emphasize the importance of regular aerobic and resistance physical activity in alleviating menopausal symptoms, improving quality of life, and reducing morbidity and mortality associated with common comorbidities.

Moreover, the introduction should include references that emphasize the relevance of the topic and identify existing gaps in the literature. For example, key conceptual definitions are only introduced later in the discussion section, which weakens the logical structure of the manuscript.

The study rationale must also be better justified. Research that investigates the impact of physical activity during menopause in greater depth is both urgent and highly relevant, especially considering the increasing life expectancy of women, growing concern with quality of life, and global population aging.

Author response: Foundational background information about the importance of physical activity for menopausal females and established recommendations for exercise were included in the introduction to contextualize and highlight the importance of this topic. Specifically, we stated: “Physical activity combats the musculoskeletal changes associated with menopause, reduces the risk of chronic disease, and improves quality of life in menopausal females. Middle-aged females are encouraged to follow established recommendations for physical activity, including both aerobic exercise and resistance training for optimal health outcomes” (lines 69-72). We believe the improvements to the Introduction draw attention to the importance of physical activity during menopause and better justify our study rationale.

2. Materials and Methods

Several methodological points need clarification:

- Exclusion criteria are not specified.

- Was surgical menopause considered in participant selection?

- Regarding the Menopause Rating Scale (MRS), it is recommended that the full 5-point Likert scale used in the study be described.

- The handling of missing data is unclear. It should be stated whether participants were required to complete all questions for the questionnaire to be considered valid.

Author response:

Exclusion criteria were added to the methods section, “Participants were excluded if they were male or if they did not meet minimum physical activity volume described in the inclusion criteria” (lines 111-112).

Surgical menopause was not considered in participant selection, but participants who reported being post-menopausal were asked if they had a natural or surgical menopause. These details were added to the methods section (lines 121-122) and the results section (lines 169-170).

The full 5-point Likert scale for the Menopausal Rating Scale was added to the methods (line 128).

A statement was added to indicate that participants were required to complete the menopausal status and Menopausal Rating Scale questions for the questionnaire to be considered valid to the methods section (lines 113-114).

3. Results

This section requires several improvements for clearer and more effective presentation of the data:

- The statement “Prevalence of menopausal symptoms among female endurance athletes is high” must be clarified or rewritten to reflect the data accurately.

- Figures and tables present overlapping information. Therefore:

- Figure 1 and Table 1 convey the same content; I recommend removing Figure 1.

- Table 1 should present results stratified by menopausal status.

- Figures 2, 3, 4, and 5 should be removed, as their data are already included in Table 2.

- Table 2 requires improvement to enhance clarity and interpretability.

- In the paragraph stating “Several menopausal symptoms were reported to negatively affect training”, I recommend summarizing only the key findings, as the detailed information is already well presented in the accompanying table.

Author response:

“Prevalence” was replaced with “frequency” throughout the manuscript in response to Reviewer 2’s comments.

Figures 1, 2, and 3 were removed as suggested to prevent overlapping information. Figures 4 and 5 were answers to the questions “In general, do you perceive that your menopausal symptoms have negatively affected your training” and “…negatively affected your performance,” and are thus separate from the information in the tables addressing individual menopausal symptoms. This was clarified in the results section (lines 207 - 211).

Table 1 was updated to present results stratified by menopausal status as suggested.

Table 2 was split into two new tables (now Table 2 – Perceived effect of menopausal symptoms on training, and Table 3 – Perceived effect of menopausal symptoms on performance), and these tables were modified to improve clarity.

The paragraph about perceived negative effects on training and performance was revised to summarize key findings to prevent repeating information from the table (lines 189-204).

4. Discussion

The discussion is overly long and lacks comparative analysis with findings from other studies involving menopausal women.

While the authors provide a detailed account of the study's limitations, the manuscript would benefit from a more balanced perspective by also highlighting the strengths of the study. This would enhance the critical value and credibility of the discussion and contribute to the interpretation of the findings in a broader scientific context.

Author response:

We describe how our results compare to rates reported of menopausal symptoms in both the general population (lines 286-288; 319-321), and where known, menopausal athletes (lines 292-297).

We also describe that the severity of menopausal symptoms in our sample is similar to those reported in the general population

---

## [Decision Letter · Decision Letter 1]

16 Oct 2025

Frequency and perceived influence of menopausal symptoms on training and performance in female endurance athletes

PONE-D-25-25663R1

Dear Dr. Hamilton,

We’re pleased to inform you that your manuscript has been judged scientifically suitable for publication and will be formally accepted for publication once it meets all outstanding technical requirements.

Kind regards,

Luiz F. Baccaro

Academic Editor

PLOS ONE

Additional Editor Comments (optional):

Reviewers' comments:

Reviewer's Responses to Questions

**Comments to the Author**

Reviewer #2: All comments have been addressed

Reviewer #3: All comments have been addressed

2. Is the manuscript technically sound, and do the data support the conclusions?

Reviewer #2: Yes

Reviewer #3: Yes

3. Has the statistical analysis been performed appropriately and rigorously?

Reviewer #2: Yes

Reviewer #3: Yes

4. Have the authors made all data underlying the findings in their manuscript fully available?

Reviewer #2: Yes

Reviewer #3: Yes

5. Is the manuscript presented in an intelligible fashion and written in standard English?

Reviewer #2: Yes

Reviewer #3: Yes

Reviewer #2: The authors met most of the reviewers' recommendations and the manuscript is in a position to be accepted.

Reviewer #3: I have reviewed the revised version of the manuscript entitled “Frequency and perceived influence of menopausal symptoms on training and performance in female endurance athletes.” The authors have addressed the previous comments thoroughly and substantially improved the quality and clarity of the paper.

The study provides valuable insights in the interaction between menopausal symptoms and athletic performance in endurance-trained women. The methodological approach is sound and the analyses are clearly presented. The discussion now provides a balanced interpretation of the findings within the existing scientific literature, and the conclusions are supported by the data.

The manuscript now meets the journal’s standards for publication. I therefore recommend acceptance for publication in its current form

**Do you want your identity to be public for this peer review?** For information about this choice, including consent withdrawal, please see our Privacy Policy

Reviewer #2: **Yes: ** Ricardo Ney Cobucci

Reviewer #3: No

---

## [Editor Report · Acceptance letter]

PONE-D-25-25663R1

PLOS ONE

Dear Dr. Hamilton,

I'm pleased to inform you that your manuscript has been deemed suitable for publication in PLOS ONE. Congratulations! Your manuscript is now being handed over to our production team.

Kind regards,

on behalf of

Dr. Luiz F. Baccaro

Academic Editor

PLOS ONE